# Characteristics of Chimeric West Nile Virus Based on the Japanese Encephalitis Virus SA14-14-2 Backbone

**DOI:** 10.3390/v13071262

**Published:** 2021-06-29

**Authors:** Guohua Li, Xianyong Meng, Zhiguang Ren, Entao Li, Feihu Yan, Jing Liu, Ying Zhang, Zhanding Cui, Yuetao Li, Hongli Jin, Zengguo Cao, Le Yi, Pei Huang, Hang Chi, Hualei Wang, Weiyang Sun, Tiecheng Wang, Yuwei Gao, Yongkun Zhao, Songtao Yang, Xianzhu Xia

**Affiliations:** 1Key Laboratory of Jilin Province for Zoonosis Prevention and Control, Changchun Veterinary Research Institute, Chinese Academy of Agricultural Sciences, Changchun 130122, China; histliguohua@163.com (G.L.); liet0706@163.com (E.L.); yanfh1990@gmail.com (F.Y.); zhangy99321@163.com (Y.Z.); Loyee1988@foxmail.com (L.Y.); ch_amms@163.com (H.C.); sunweiyang1987@163.com (W.S.); wgcha@163.com (T.W.); gaoyuwei@gmail.com (Y.G.); 2College of Animal Science and Technology, Shihezi University, Shihezi 832003, China; 3College of Animal Science and Technology, Jilin Agricutural University, Changchun 130118, China; xianyongmeng163@163.com (X.M.); tata989837@gmail.com (Z.C.); 4School of Basic Medical Sciences, Henan University, Kaifeng 475004, China; renzhiguang66@126.com (Z.R.); 15294861003@163.com (J.L.); 5College of Animal Science and Veterinary Medicine, Henan Institute of Science and Technology, Xinxiang 453003, China; liyuetao@hist.edu.cn; 6Key Laboratory of Zoonosis Research, Ministry of Education, College of Veterinary Medicine, Jilin University, Changchun 130062, China; jin8616771@163.com (H.J.); cengcao@hotmail.com (Z.C.); hp19932015@163.com (P.H.); whl831125@163.com (H.W.)

**Keywords:** Japanese encephalitis virus, West Nile virus, chimera

## Abstract

West Nile virus disease (WND) is an arthropod-borne zoonosis responsible for nonspecific fever or severe encephalitis. The pathogen is West Nile virus belonging to the genus *Flavivirus,* family *Flaviviridae*. Every year, thousands of cases were reported, which poses significant public health risk. Here, we constructed a West Nile virus chimera, ChiVax-WN01, by replacing the *prMΔE* gene of JEV SA14-14-2 with that of the West Nile virus NY99. The ChiVax-WN01 chimera showed clear, different characters compared with that of JEV SA14-14-2 and WNV NY99 strain. An animal study indicated that the ChiVax-WN01 chimera presented moderate safety and immunogenicity for 4-week female BALB/c mice.

## 1. Introduction

West Nile virus disease is a wide-spread arboviral zoonosis. The clinical manifestation varies from nonspecific febrile illness and can progress to more severe hemorrhagic fever or encephalitis [1]. The pathogen of WND is West Nile virus (WNV), which was first isolated from the blood of a woman in Uganda in 1937 [2]. West Nile virus disease was sporadic in the areas of Africa, Europe, and west Asia, at first. But later, it spread to the United States in 1999 [3]. According to statistics from the American Animal and Plant Health Inspection Service (APHIS), over 27,000 cases of horse infections were diagnosed in the United States up to October, 2020 [4]. From 1999–2019, there were 51,801 reported human cases, with 2330 (5%) cases of death and 25,290 cases of neuroinvasive disease [5]. There is no special treatment for the WND, and vaccine prevention is one of the most effective measures.

WNV, belonging to genus *Flavivirus*, family *Flaviviridae*, is a single-stranded, positive RNA virus with a genome of about 11,000 nt. The genome encodes a polyprotein that can be further cut into three structural proteins (C, prM, E) and seven nonstructural proteins (NS1, NS2A, NS2B, NS3, NS4A, NS4B, NS5). There are many types of flavivirus, which include Japanese encephalitis virus (JEV), yellow fever virus (YFV), dengue virus (DENV), zika virus (ZIKV), and tick-borne encephalitis virus (TBEV) [6]. The clinical manifestations of infection vary considerably from nonspecific febrile illness to more severe syndromes, including hemorrhagic fever, encephalitis, or microcephaly. Among them, West Nile fever (WNF) and West Nile encephalitis (WNE) are the main clinical syndromes of WNV infection.

Researchers have taken plenty of measures to develop the vaccine candidates. There are currently four equine West Nile virus vaccines licensed. Three of them are inactivated vaccines, and another one is a live canarypox virus vaccine containing the West Nile virus structural gene (the live vaccine cannot be replicated in the body) [7]. Although there are good immune protection, horses are needed a booster once a year for sustained immune protection after the first immunization, and the cost of vaccines and labor will increase accordingly.

Due to the better safety and protection, two licensed live-attenuated vaccines (YFV 17D and JEV SA14-14-2) have been used as vectors to construct recombinant flavivirus. Arroyo (2004) developed the ChimericVax-WN_02_ vaccine candidate; after phase I [8] and phase II clinical trials [9,10], it was safe and effective, eliciting higher seroconversion rates after a single dose than those reported vaccine candidates [11]. Li XF (2013) developed a chimeric virus called ChinWNV, which replaced the *prMΔE* gene of JEV SA14-14-2 backbone with that of WNV strain Chin-01. A single dose of ChinWNV inoculation induced robust immune responses and conferred significant protection against lethal WNV challenge. Additionally, the neurovirulence and neuroinvasiveness were attenuated [12]. But the strain Chin-01 is not the most prevalent one in the world. Other vaccine candidates include inactivated vaccines developed by Barrett (2017) and Woods (2019), which also presented antibody response [13,14]; rWN/DEN4Δ30 vaccine candidates based on DENV4 backbone presented high levels of neutralizing antibody after phase I clinical trials by Durbin (2013) [15] and Pierce (2017) [16]; DNA vaccine candidates were tested to have neutralizing antibody titers, but there were no further evaluations. 

Most of the chimeric viruses were constructed based on the vaccine strain YFV 17D backbone due to the lowered virulence. However, compared with JEV SA14-14-2, the neurovirulence of YFV 17D is stronger [11,17,18]. JEV SA14-14-2 strain has been a successful, licensed, live vaccine used in China since 1989 for Japanese encephalitis prevention [19]. Considering the lower neurovirulence of JEV SA14-14-2 and lower side effects, we employed JEV SA14-14-2 as a backbone to construct chimeric virus for further viral research. WNV is required to be cultivated in the laboratory with the biosafety level 3 (BSL-3), which is a bottleneck for research. And there is no better live-attenuated WNV vaccine licensed. Considering the manipulation of WNV-related research at BSL-2 laboratory and exploration of chimeric WNV characters for vaccine development, we constructed the chimeric WNV, named ChiVax-WN01.

Although there are several chimeras in trial, which one will eventually be the best is uncertain. The variety of chimeric virus research may confer many more vaccine reservoirs.

## 2. Materials and Methods

### 2.1. Cells and Viruses

BHK-21 (baby hamster kidney) and BSR T7/5 (derived from BHK-21 that can express T7 polymerase) cells were maintained in the Dulbecco’s minimal essential medium (DMEM) (Corning, Corning, NY, USA) containing 10% heat-inactivated foetal bovine serum (FBS, Biological Industries, Israel) and 1% penicillin-streptomycin (Caisson, Smithfield, UT, USA). All cells were incubated at 37 °C in 5% CO_2_. JEV SA14-14-2 strain (GenBank accession no. MK585066) was prepared and titrated by BHK-21 cells. The following chimeric virus was recovered in the BSR T/7 cells and titrated by BHK-21 cells.

### 2.2. Chimeric Plasmid Construction

A previously described, the JEV SA14-14-2 infectious clone we built, pFLJEV, [20] was used to construct the chimeric infectious clone. The structural gene (prMΔE) of WNV NY99 strain (GenBank accession no. DQ211652) was synthesized by Sangon Biotech Co., Ltd. (Shanghai, China). The prMΔE genes of WNV NY99 were engineered to permit the replacement of the corresponding sequence of JEV SA14-14-2 (Figure 1a) according to the polyprotein cleavage sites in Appendix A [12,21]. The fragment KLM was amplified by fusion PCR with fragments K, L, and M (Figure 1b). Fragments K, L, and M were amplified, respectively, by the following primers (Table 1) with JEV SA14-14-2 cDNA (fragment K and M) or WNV prME DNA (fragment L) as templates. The detailed amplification protocols are listed in Appendix A. Fragments KL and KLM were amplified by the fusion PCR (Appendix A). The pACYC177-linker is constructed in Appendix A. The subclone (pACYC177-KLM) (Appendix A) and the full-length infectious clone (pChiVax-WN01) (Appendix A) were constructed by the molecular biology protocols. All the plasmids were confirmed by DNA sequencing.

### 2.3. Recovery of Chimeric Virus ChiVax-WN01

The BSR/T7 cells were prepared in 60% confluence in 6-well plate. A total of 2.5 μg of the chimeric West Nile virus infectious clones pChiVax-WN01 were directly transfected into the BSR/T7 cells grown in one well of 6-well plate with Lipofectamine 3000 (Thermo Scientific, Waltham, MA, USA) according to the transfection protocols. The transfected cells were successively observed 4–6 days at 37 °C in 5% CO_2_ for virus recovery. A total of 200 μL cells culture supernatants were replaced with fresh 10% FBS DMEM every 24 h for maintaining the cells’ viability. The suspension of cells and supernatant was harvested by freeze-thaw once at −80 °C or −20 °C when the transfected BSR/T7 cells were observed to have typical cytopathic effects (CPE). The progeny chimeric virus suspensions (named ChiVax-WN01) from the transfection step were proliferated serially into 4 passages in the BSR/T7 cell line to produce P5 virus and were stored in aliquots at −80 °C until use. Virus titers were determined by standard plaque assay on BHK-21 cells.

### 2.4. Plaque Assay

BHK-21 cells were plated into 12-well plate with 5 × 10^5^ cells/well and cultured for 24 h at 37 °C in CO_2_. P5 ChiVax-WN01 were serially diluted by ten-fold (range from 10^−1^–10^−5^, with 3 repeats each dilution) with DMEM. The diluted viruses were inoculated, respectively, to the plated cells and adsorbed for 1 h at 37 °C with gentle rocking every 15 min. After adsorption, the virus was discarded, and the cells were rinsed with PBS or DMEM. The cells were overlaid with 1 mL nutrient agarose (1:1 of 3% low-melting-point agarose and 2 × DMEM with 20% FBS and 2% penicillin-streptomycin) per well. After being incubated at 37 °C and 5% CO_2_ for 3–5 days, cells were fixed with 500 μL/well of 10% formaldehyde and stained with 300 μL/well crystal violet solution (Beyotime, Shanghai, China) to visualize the plaques.

### 2.5. Growth Curves

Growth curve was measured with BHK-21 cells in 12-well plate. The P5 ChiVax-WN01 chimera was inoculated into 4 wells (5 × 10^5^ cells/well) of BHK-21 cells, respectively, at a multiplicity of infection (MOI) of 0.01. The viruses were discarded after adsorption for 1 h at 37 °C and the cells were rinsed with PBS three times. Then, the culture medium was added into each well. The suspension of cells and viruses were collected at 24 h intervals (24 h, 48 h, 72 h, and 96 h) post inoculation and stored at −80°C. Virus titers at different time points were measured by plaque assay mentioned above.

### 2.6. Morphology

The ChiVax-WN01 chimera was observed under a transmission electron microscope (TEM) HITACHI H7650 (Hitachi, Tokyo, Japan). A total of 2 × 10^6^ BHK-21 cells in a T25 flask were inoculated with the ChiVax-WN01 chimera at a MOI of 0.01 and incubated at 37 °C 5% CO_2_. The culture supernatants were discarded, and the cells were collected by scratching or trypsin-digestion after 48 h post infection. The collected cells were resuspended with 1 mL 10% FBS DMEM culture and centrifuged for 1 min at 10,000 rpm to obtain the pellet. The pellet was processed according to the ultrathin sectioning protocol [22] for TEM observation.

### 2.7. Genetic Stability Assay

The chimeric viral RNA of P5 and P10 ChiVax-WN01 was extracted according to the viral RNA extraction kit protocol (Bioflux, Hangzhou, China), and cDNA was synthesized and sequenced as Li XF described [21].

### 2.8. Virulence in Mice

Four-week-old female BALB/c mice were used in this study. For the neuroinvasive study, groups of mice (*n* = 8–10) were inoculated intraperitoneally (i.p.) with ten-fold dilution (10^5^, 10^4^, 10^3^, 10^2^, and 10^1^ PFU) of ChiVax-WN01, 10^5^ PFU of JEV SA14-14-2, and PBS, respectively. For the neurovirulence study, groups of mice (*n* = 10) were inoculated intracranially (i.c.) with 100 PFU of ChiVax-WN01, 100 PFU of JEV SA14-14-2, and PBS. Mice survival was monitored 15 days post inoculation. 

### 2.9. Assessment of Virus Distributions in Infected Mice

Groups (*n* = 15) of 4-week female BALB/c mice were inoculated intraperitoneally with 100 PFU JEV SA14-14-2, 100 PFU ChiVax-WN01, and 100 μL PBS, respectively. Mice were collected for the plasmas, brains, livers, spleens, lungs, and kidneys on day 1, 3, 5, 7, and 9 to detect the virus titers according to McAuley [23].

### 2.10. Microneutralization Assay

Groups of 4-week-old female BALB/c mice (*n* = 10) were inoculated intraperitoneally with 100 PFU of JEV SA14-14-2, ChiVax-WN01, and PBS, respectively. The 100 PFU of ChiVax-WN01 i.c. could cause mice death, so 100 PFU was employed. The sera were collected at 3- and 5-weeks post inoculation for measuring the neutralizing antibody titers. Neutralizing antibody titers were determined by microneutralization test. Briefly, sera were serially 2-fold diluted from 2^−3^ to 2^−8^ in the 96-well plates. The 100 PFU of ChiVax-WN01 were inoculated into the wells to mix with the diluted sera and incubated 1 h at 37 °C. BHK-21 with 3000 cells per well were seeded into the virus-sera mixture in the 96-well plate and were incubated at 37 °C 5% CO_2_ for 4–5 days post infection. Finally, the cells were monitored the CPE.

### 2.11. ELISpot IFN-γ, IL-2, and IL-4 Assay

Three groups of mice (*n* = 15) were inoculated intraperitoneally with 100 PFU JEV SA14-14-2, 100 PFU ChiVax-WN01, and PBS, respectively. Mice splenocytes were collected and seeded into 96-well ELISpot plate (MABTECH, Nacka, Sweden) after 15 weeks post inoculation for ELISpot IFN-γ, IL-2, and IL-4 assay as described previously [24,25]. Splenocytes were restimulated with E80 (truncated envelope protein of WNV NY99) at a final concentration of 10 μg/mL or 10^3^ PFU ChiVax-WN01 per well for cytokine production. After being incubated at 37 °C 5% CO_2_ for 48 h, splenocytes secreting IFN-γ, IL-2, and IL-4 were detected by mouse enzyme-linked immunospot (ELISpot) kit according to the protocols. The spot-forming cells (SFCs) were counted by the ELISpot reader (Multispotreader Spectrum, AID, Strasberg, Germany).

### 2.12. Statistical Analysis

Statistical analyses of growth curves, neutralizing antibody titer, and survival curves were drawn by the GraphPad Prism software, version 6.0. Significance differences were analyzed by one-way ANOVA.

## 3. Results

### 3.1. Chimeric Plasmid Construction

In order to acquire the chimeric infectious clone, *prMΔE* gene of JEV SA14-14-2 was replaced with that of the WNV NY99 based on the JEV SA14-14-2 infectious clone pFLJEV. The constructed chimeric infectious clone containing the *prM**Δ**E* gene of WNV NY99 was named pChiVax-WN01. The pChiVax-WN01 contained the prM signal peptide sequence of JEV and the last three amino acids at the end of envelope protein to maintain efficient cleavage at the junction of C-prM and E-NS1. The constructing strategy is shown in Figure 1. In order to verify the accuracy of the genomic sequence of the chimeric plasmid, pChiVax-WN01 was sequenced by Sanger DNA sequencing and was aligned with that of JEV SA14-14-4 (GenBank accession no. MK585066) and that of WNV NY99 (GenBank accession no. DQ211652). It indicated that pChiVax-WN01 had consensus sequence except for two silent mutations at WNV T2250C and JEV A9143G.

### 3.2. Chimeric Virus Recovery, CPE, and Virus Stability

To verify the recovery efficiency of pChiVax-WN01, 2.5 μg pChiVax-WN01 were directly transfected into the BSR T7/5 cells with lipofectamine 3000 according to the protocol. The result showed that clear CPE appeared at 81–96 h post transfection, whereas there was no CPE in the Mock (Figure 2). The harvested chimeric virus was named P1 ChiVax-WN01. In order to improve the viral titer, ChiVax-WN01 were propagated serially with four passages (named P5 ChiVax-WN01) in BSR T7/5 cells, and the titer of P5 ChiVax-WN01 reached 2 × 10^6^ PFU/mL. In order to verify the genomic stability of the chimeric virus, the complete sequence of P5 ChiVax-WN01 were sequenced and aligned with that of JEV SA14-14-2 and WNV NY99. The results presented that all the genomic sequence had consensus sequence.

### 3.3. Plaque Morphology

In order to observe the plaque sizes of the P5 ChiVax-WN01 and that of JEV SA14-14-2, both viruses were cultured by plaque assay. The results showed that the plaque sizes of ChiVax-WN01 were 1.91 ± 0.24 mm in diameter (Figure 3a), which were much bigger than that of the JEV SA14-14-2, with plaque size of 0.82 ± 0.08 mm in diameter (Figure 3a), and were much smaller than that of the WNV NY99 plaques (3.60 ± 0.52 mm) [23]. It indicated that bigger plaques of viruses have positive correlation with higher neurovirulence among the three flaviviruses.

### 3.4. Growth Curves

In order to assess the growth efficiency of ChiVax-WN01, the growth curves of ChiVax-WN01 and JEV SA14-14-2 were measured according to the plaque assay protocols. The results showed that ChiVax-WN01 reached peak titer at 48–72 h post inoculation (p.i.), which was lower than that of JEV SA14-14-2 at the same time points (Figure 3b). There were significant differences in the viral titers between JEV SA14-14-2 and ChiVax-WN01 at 48 h, 72 h, and 96 h. The results suggested that ChiVax-WN01 has a weaker growth ability.

### 3.5. Virion Morphology

In order to identify if there are virion particle formations, BHK-21 were inoculated with ChiVax-WN01 at a MOI of 0.01 and harvested for TEM observation. The results indicated that there were round virions. The size of a single virion is around 40 nm in diameter (Figure 3c).

### 3.6. ChiVax-WN01 Virulence in Mice

In order to assess the neuroinvasiveness of ChiVax-WN01 in mice, groups of mice were inoculated i.p. with serial ten-fold dilutions of ChiVax-WN01 or JEV SA14-14-2 and were observed for clinical manifestations and death for 15 days. Groups of mice infected with 10^5^–10^1^ PFU of ChiVax-WN01 and mice infected with 10^5^ PFU of JEV SA14-14-2 had no death or encephalitis-related symptoms (Appendix A). Both ChiVax-WN01 and JEV SA14-14-2 had less neuroinvasiveness than that of WNV NY99 [23].

In order to assess the neurovirulence of ChiVax-WN01, groups of mice were inoculated i.c. with 100 PFU of ChiVax-WN01 [23] or 100 PFU of JEV SA14-14-2. The results showed that group inoculated i.c. with ChiVax-WN01 had 5 occurrences of death (50% mortality) during days 7–10 post infection, whereas no death was observed in the JEV SA14-14-2 group i.c. (Figure 4a). It indicated that ChiVax-WN01 presented higher neurovirulence than that of JEV SA14-14-2. Groups of mice inoculated i.p. with 100 PFU of ChiVax-WN01 showed no neuroinvasiveness (Figure 4b). Groups of mice inoculated i.c. or i.p. with parental JEV SA14-14-2 manifested no death or signs of illness in 15 days post infection (Figure 4c). It indicated that JEV SA14-14-2 had no obvious neurovirulence and neuroinvasiveness. According to the results above, the neurovirulence of ChiVax-WN01 was decreased obviously compared with that of the WNV NY99 that McAuley measured [23] but increased compared with that of JEV SA14-14-2; the neuroinvasiveness of ChiVax-WN01 was decreased greatly compared with that of WNV NY99 [23] and not increased compared with that of JEV SA14-14-2.

### 3.7. Virus Distributions in Mice

In order to detect virus proliferation in the blood and viscera, three mice in each group were identified for measurement of the viral titers in the plasmas, brains, livers, spleens, lungs, and kidneys by plaque assays on day 1,3, 5, 7, and 9, respectively. The results indicated that there were no viral titers detected in the plasma and viscera (Appendix A).

### 3.8. Neutralizing Antibody Titers

To compare the difference of neutralizing antibodies induced by the ChiVax-WN01 and JEV SA14-14-2, neutralizing antibody titers were tested. Although plaque reduction neutralization test (PRNT) is the gold standard for neutralizing antibody assay, it is time-consuming and laborious. There is correlation of results between PRNT and virus neutralization assay (VNA) [26]. We employed ChiVax-WN01 to detect the neutralizing antibody titer of mice by VNA. The results showed that the neutralizing antibody titer in the ChiVax-WN01 group and the JEV SA14-14-2 group was more than 1:10. In general, a neutralizing antibody titer over 1:10 is recognized as an indicator of protection [12]. There were significant differences between the ChiVax-WN01 group and Mock group at 21 days and 35 days post infection; however, there were no significant differences between the JEV SA14-14-2 and Mock group (Figure 5). These results suggested that ChiVax-WN01 could induce stronger neutralizing antibody titers than JEV SA14-14-2 and Mock.

### 3.9. Cytokine Secretion by Restimulated Splenocytes

In order to assess the immune response, the mice were checked for detection of the splenocytes secreting IFN-γ, IL-2, and IL-4, and the spots were counted by the ELISpot Reader. The results showed that the number of splenocytes secreting Th1 cytokines (IFN-γ and IL-2) in the ChiVax-WN01 and JEV SA14-14-2 group presented significantly higher than that of the Mock group. However, that of Th2 cytokine (IL-2) in both viruses groups had no significant difference compared with the Mock group (Figure 6). The results suggested that ChiVax-WN01 and JEV SA14-14-2 induce stronger cellular immune responses and weak humoral immune responses.

## 4. Discussion

West Nile virus disease (WND) is a zoonosis with the main clinical symptoms of West Nile fever (WNF) or West Nile encephalitis (WNE). Although there are four licensed horse vaccines used in the market, they need booster inoculation every year after primary immunization, which increases the costs.

In this study, we constructed a live chimeric WNV, named ChiVax-WN01, based on the JEV SA14-14-2 infectious clone. It presented clear biological differences JEV SA14-14-2 and WNV NY99. Mice study showed that ChiVax-WN01 exhibited better safety with regard to the attenuated neuroinvasiveness and could induce humoral and cellular immune response.

JEV and WNV are positive, single-stranded RNA viruses with only an open reading frame to translate the genome into a polyprotein. The correct proteolytic processing of polyproteins is of great importance to the virion assembly. Virus-encoded serine protease, NS2B-3, processes at the C/prM signal peptide junction. Host signal peptidase cleaves between C/prM, prM/E, and E/NS1 conjunctions [6]. In this study, we used the *prMΔE* of WNV NY99 to replace that of JEV SA14-14-2 but preserved the prM signal peptide and the last three amino acids (VHA) of envelope protein (E) of the JEV SA14-14-2 backbone for precise cleavage. The last one and the last reciprocal third amino acids of JEV E are conserved among many flaviviruses [22]. Chimeric ChiVax-WN01 could be successfully recovered by this strategy.

pChiVax-WN01 with the equal microgram of pFLJEV led to the cytopathic effect (CPE) at 81 h post transfection. The appearance of CPE was earlier than that of the JEV SA14-14-2 infectious clone, pFLJEV, which showed clear CPE at 96 h post infection. It seems that pChiVax-WN01 presents stronger CPE. The plaque sizes of JEV SA14-14-2 (0.82 ± 0.08 mm), ChiVax-WN01 (1.91 ± 0.24 mm), and WNV NY99 (3.60 ± 0.52 mm) were increased and accompanied by increase of the neurovirulence. This increase in plaque size translated into an increase in neurovirulence. Likewise, smaller plaques are often considered proxy of attenuation and vaccine safety. On the other hand, introduction of WNV *prMΔE* led to the ChiVax-WN01 neurovirulence increase (compared with that of JEV SA14-14-2), which also reflected that the structural genes (*prME*) are closely related to neurovirulence [11]. According to the above results, we deduced that the structural gene *prMΔE* influences viral plaque size and neurovirulence, and the backbone (JEV SA14-14-2 without *prMΔE*) might contribute to the neuroinvasiveness.

We constructed two kinds of pChiVax-WN01 infectious clones (one with introns and the other with no intron). The pChiVax-WN01 with introns had not recovered a progeny virus after several transfection and blind passages; however, the pChiVax-WN01 without intron successfully rescued a progeny virus post transfection. It seems that the existence of introns interfered with the virus recovery. However, discard of introns is hard to manifest in an infectious clone. We changed the construction strategy to ligate the KLM (Figure 1b) first into the vector pACYC177-linker then the following sequence of JEV SA14-14-2 (F345-HDVr) was ligated into pACYC177-KLM, and the E. coli HB101 were cultured at room temperature, which helped the pChiVax-WN01 infectious clone construction.

There are mainly two ways of rescuing virus: one is in-vitro transcription of viral RNA and transfection of the transcribed viral RNA; the other is direct transfection of the viral infectious clone plasmid. Arroyo used the two-plasmid system to construct infectious DNA clone and transcribed it in vitro and transfected to rescue the progeny virus [11]. Li XF recovered chimeric ChinWNV by the former way as well [12]. Transfection of in-vitro-transcribed viral RNA could control and accelerate the viral recovery process. Generally, there is high viral titer after several days post transfection. However, in-vitro-transcribed viral RNA are easily degraded in room conditions, and they need to be stored at −80 °C, which is less convenient. In this study, we directly transfected the infectious clone plasmid pChiVax-WN01 to BSR T7/5 cells to get the viral RNA, which shortened the viral recovery process and was done easily in the normal conditions. However, the first passage of the progeny virus had a low viral titer due to the amount of viral RNA transcripts that could not be controlled in the cells. Both viral rescuing ways have their advantages and disadvantages.

ChiVax-WN01 virulence has been tested by neuroinvasiveness, neurovirulence, and virus distribution assays in mice. In this study, mice inoculated intraperitoneally with different titers (10^5^–10^1^ PFU) of ChiVax-WN01 all survived, with no difference from that of JEV SA14-14-2, indicating that ChiVax-WN01 lacks neuroinvasiveness. Out of the mice inoculated intracranially with 100 PFU ChiVax-WN01, half survived, with significant difference from JEV SA14-14-2 and PBS groups, indicating that ChiVax-WN01 had neurovirulence, and a further measure should be taken to decrease the neurovirulence. Mice inoculated i.c. with 100 PFU WNV NY99 all died [23]. However, compared with that of WNV NY99, the neurovirulence and the neuroinvasiveness of ChiVax-WN01 were attenuated greatly. There was no virus detected in the blood and viscera of intraperitoneally immunized mice at different time points, which showed low or no viral proliferation, indicating higher safety of ChiVax-WN01. Mice that succumbed to i.c. inoculation may produce adaptive mutation by the immune system that combats the chimera; therefore, it is necessary to try to isolate the virus from succumbed mice brains for further research.

Ideally, the challenge study should have been used to measure the efficacy of ChiVax-WN01; however, we have no BSL-3 facilities for WNV research. We will carry out the challenge study and compare the efficacy of the chimeric viruses after constructing 2-3 WNV chimera.

In all, we constructed a chimeric WNV virus named ChiVax-WN01. It was attenuated greatly compared with that of wild-type WNV NY99, although it contains neurovirulence. Therefore, ChiVax-WN01 may be not a suitable vaccine candidate, but attenuation of the neurovirulence will help. Another neurovirulence-attenuated strain of ChiVax-WN01was constructed, and the characteristics will be discussed in another article. This ChiVax-WN01 could be safely manipulated at BSL-2 for research and has laid the foundation for the development of a much safer, live-attenuated vaccine candidate.

## Figures and Tables

**Figure 1 viruses-13-01262-f001:**
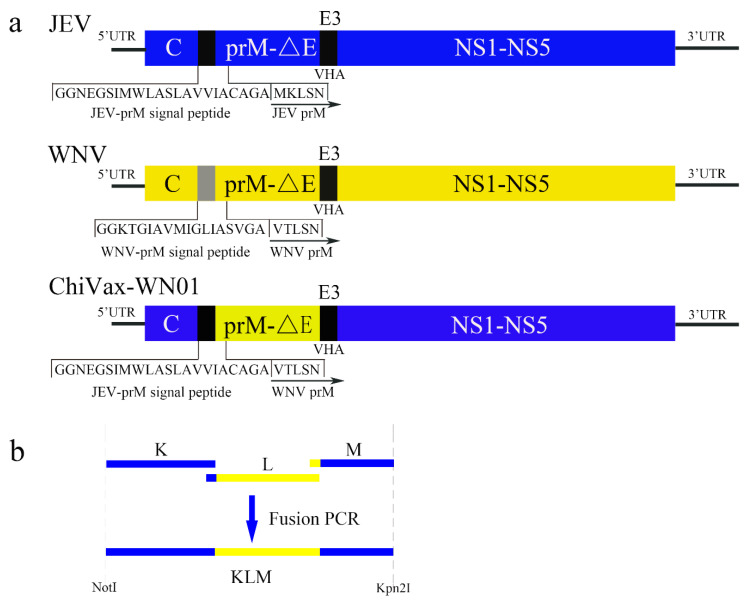
Schematic diagram for the construction of chimeric ChiVax-WN01. (**a**) Replacement of WNV prMΔE with that of JEV SA14-14-2: ΔE represented the E protein without the last three amino acids at the carboxyl terminus. E3 referred to the last three amino acids of E protein. (**b**) Introduction of WNV *prMΔE* by fusion PCR of fragments K, L, and M.

**Figure 2 viruses-13-01262-f002:**
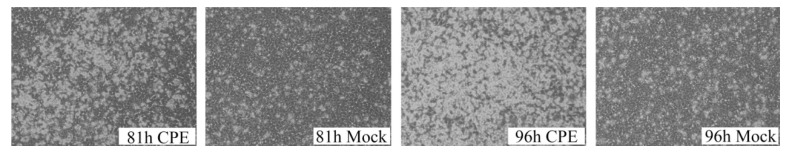
CPE post transfection (100×). CPE of BSR T7/5 cells post transfection at 81 h and 96 h, respectively. White pyknotic cells appeared at 81 h after transfection and spread to cover almost the whole well at 96 h after transfection, whereas cells in the Mock remained normal.

**Figure 3 viruses-13-01262-f003:**
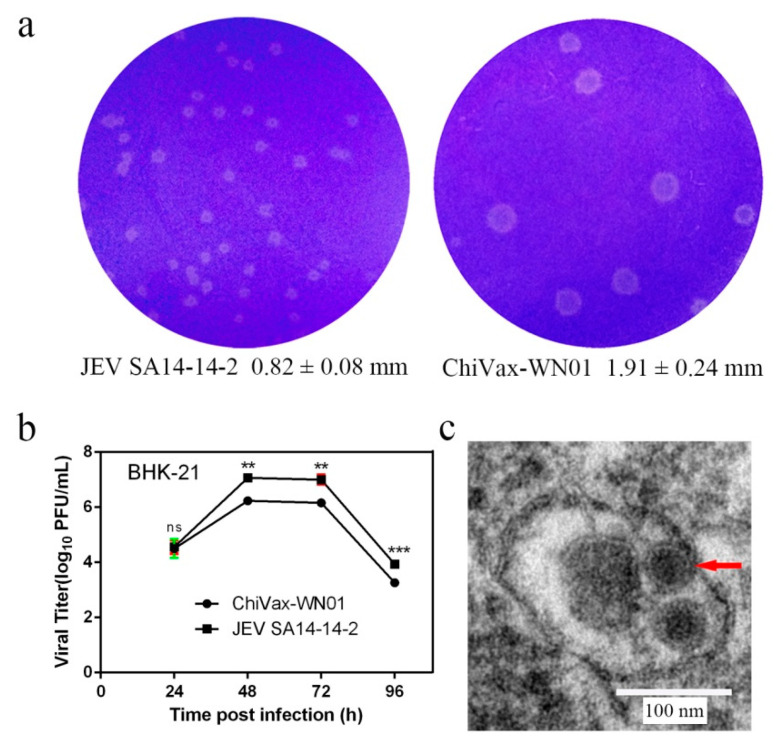
Characteristics of ChiVax-WN01. (**a**) Plaque morphology and size of JEV SA14-14-2 and ChiVax-WN01 in BHK-21 cells at 5 days post inoculation (d.p.i.). (**b**) Growth curves of JEV SA14-14-2 and ChiVax-WN01 in BHK-21 cells. Data are shown as the means ± SDs and were analyzed by student’s *t*-test (** *p* < 0.01, *** *p* < 0.001). (**c**) Morphology of ChiVax-WN01 (red arrow) by the transmission electron microscope.

**Figure 4 viruses-13-01262-f004:**
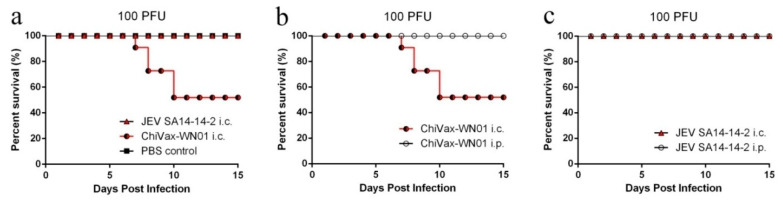
Virulence of viruses following intraperitoneal (i.p.) and intracranial (i.c.) inoculation. (**a**) Four-week-old BALB/c mice (*n* = 10) were inoculated with 100 PFU of ChiVax-WN01 or JEV SA14-14-2 i.c. (**b**) BALB/c mice (*n* = 10) were inoculated with 100 PFU of ChiVax-WN01 i.c. or i.p., respectively. (**c**) BALB/c mice (*n* = 10) were inoculated with 100 PFU of JEV SA14-14-2 i.c. or i.p., respectively. The ChiVax-WN01 presented clear neurovirulence but no neuroinvasiveness for mice.

**Figure 5 viruses-13-01262-f005:**
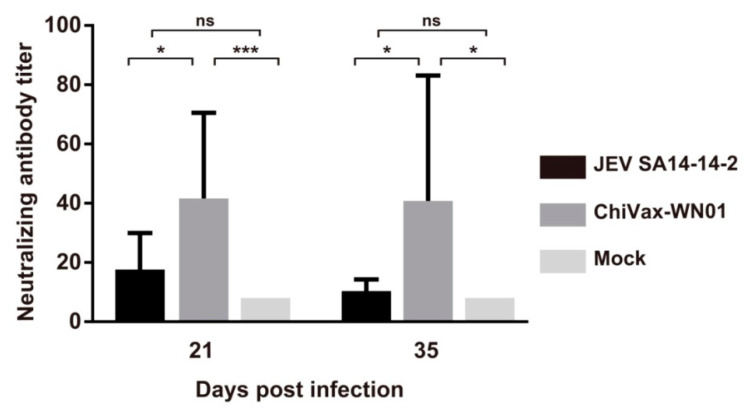
Neutralizing antibody titer. Four-week-old BALB/c mice (*n* = 10) were inoculated with JEV SA14-14-2 or ChiVax-WN01 at 100 PFU i.p., respectively. The blood was collected 21 d and 35 d post infection for detecting antibody titers by viral neutralizing assay (VNA). Data are shown as the means ± SDs and were analyzed by one-way ANOVA (* *p* < 0.05, *** *p* < 0.001).

**Figure 6 viruses-13-01262-f006:**
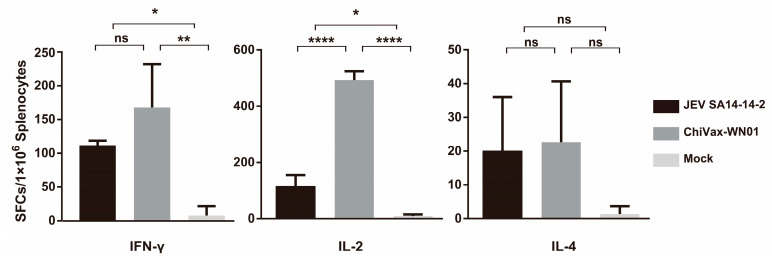
ELISpot analysis of splenocytes secreting IFN-γ, IL-2, and IL-4 from the immunized mice. Four-week-old female BALB/c mice (*n* = 3) were inoculated with JEV SA14-14-2 or ChiVax-WN01 at 100 PFU i.p. Mice splenocytes were collected after 15 weeks post inoculation and were restimulated with ChiVax-WN01 or E80 (10 μg/mL). The splenocytes secreting the IFN-γ, IL-2, and IL-4 were quantified according to the ELISpot assay. Data shown are as the means ± SD and were analyzed by one-way ANOVA (* *p* < 0.05; ** *p* < 0.01; **** *p* < 0.0001). ns indicates no significance.

**Table 1 viruses-13-01262-t001:** Primer sequences.

Primer Names	Primer Sequences 5′→3′	bp
K-F	TCTGCGGCCGCTAATACGAC	505
K-R	GGCTCCTGCACAAGCTATGACA	
L-F	TGTCATAGCTTGTGCAGGAGCCGTTACCCTCTCTAACTTCCAAGGG	2017
L-R	GTTCACGGAGAGGAAGAGCAGA	
M-F	TTCTGCTCTTCCTCTCCGTGAACGTGCATGCTGACACTGGATGTG	1008
M-R	CTGTCCGGAATCGTAGGGGC	

F indicates forward primer, and R indicates reverse primer. Restriction endonuclease sites GCGGCCGC (*Not* I) and TCCGGA (*Kpn*2 I) are underlined.

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
