# Peer review of "Characteristics of Chimeric West Nile Virus Based on the Japanese Encephalitis Virus SA14-14-2 Backbone"

_viruses, 2021, doi:10.3390/v13071262_

Round 1

Reviewer 1 Report

Reviewer:  The manuscript from Guohua Li et al. describes the construction and characteristics of a chimeric WNV using JE as a backbone.

Unfortunately the manuscript is lacking page and line numbering, making it complicated to review.

In the abstract the authors claim that they have a vaccine candidate but the manuscript is ending with that it is not a suitable candidate. They referred to another more promising WNV candidate which they have constructed but are not showing results in that manuscript. So what?

Major points:

Reviewer: The manuscript is generally lacking detailed information on the methods used. No information is for example given for the cloning procedure like Oligonucleotide sequences etc.

The gene synthesized by the company, the generation of the fragments K, L and M is not sufficiently described. This must be revised in general to allow reproduction of the experiments by readers.

Please add experimental details to all method sections to make the paper more interesting to the readers.

Minor points:

Reviewer: “structural gene” should be prME envelope sequence

Reviewer: Fig 1a: in WNV the prM start is VTLSN in ChiVax-WN01 it is VTLAA is this a typing error or a result of the cloning procedure? Please specify.

Reviewer: What is the reason for the complicated prME nomenclature since E3 is always VHA? I would skip the deltaE expression

Reviewer: Fig1b: Giving restriction sites is useless when the cloning procedure is not explained in detail in the Mat&Met section.

Reviewer: If you give the position of your silent mutation in WNV-2 and JEV-9 please add the sequence changes like X250Y, X143Y. In that regard what means “standard”? Which sequence is meant as a alignment control? Please specify.

Reviewer: In Mat&Met as well in the result section the manuscript is often using lab slang explanations. For example:

….both viruses were performed plaque assay.

…higher than that of Mock group.

Reviewer 2 Report

In this study the authors have generated a chimeric virus by substituting prM-E protein from the WNV in to JEV SA14 vaccine backbone. The growth kinetics and immune response to the chimeric virus is also characterized and compared to the parental SA14 strain.

This is an interesting study and in general the experiments are appropriately performed with some caveats to the interpretation of results and over claims.

Specific comments

Since neutralization and T cell activation assays were done against the vaccine candidate itself, it is difficult to ascertain the effectiveness of this chimeric virus as a vaccine candidate against WNV. To satisfy this claim challenge studies are needed.

Abstract:  “The chimera could be employed as a vaccine candidate after a following neurovirulence attenuation. This ChiVax-WN01 laid the foundation for the development of much safer live-attenuated vaccine candidate and could be safely manipulated at BSL-2 for research.” This is a discussion point, not supported by the data and should be removed from the abstract.

Figure.4. The chimeric virus when given intra-cranially (i.c.) led to enhanced mortality (>50%) in mice compared to its backbone SA14 strain, suggesting a gain of function phenotype. The larger plaque size in fig.3 also indicates gain of virulence.

Have the authors tried vaccinating mice peripherally followed by a high dose i.c. challenge to see if vaccination provided protection against neurovirulence? In the absence of BSL3 facility, this experiment could be performed to demonstrate vaccine-induced protection against neurovirulence.

Did authors look at the adaptive mutations in virus that may have occurred in 50% of mice those who succumbed to i.c. inoculation ?

Round 2

Reviewer 1 Report

The authors have answered all questions and I believe the manuscript has been sufficiently improved.

I only have one remark. Is the Figure R1 only for the eye of the reviewer or was it designed for the manuscript also? Figure R1 can be added easily to the Supplementary section with a reference to figure R1 in section 2.2.

Author Response

The authors have answered all questions and I believe the manuscript has been sufficiently improved.

I only have one remark. Is the Figure R1 only for the eye of the reviewer or was it designed for the manuscript also? Figure R1 can be added easily to the Supplementary section with a reference to figure R1 in section 2.2.

Response: Thank you very much for your kind advice. Due to Figure R1 had been published by Li Xiaofeng et. al., I just planned to cite his article to address the origin of chimera at first. As you suggested, to listed Figure R1 to the supplementary section is easier and will be convenient for readers. I had added the Figure R1 to the Supplementary section Figure S1.